# How Are Women Politicians Treated in the Press? The Case of Spain, France and the United Kingdom

Cristina Fernández-Rovira [1,*] and Santiago Giraldo-Luque [2]

1   Department of Communication, University of Vic—Central University of Catalonia, 08500 Barcelona, Spain
2   Department of Journalism and Communication Sciences, Autonomous University of Barcelona,
    08193 Barcelona, Spain; santiago.giraldo@uab.cat
*   Correspondence: cristina.fernandez1@uvic.cat

**Abstract:** Women politicians have been discriminated against or negatively valued under stereotypes in media coverage and have been given a secondary role compared to male politicians. The article proposes an analysis of the treatment given by digital media to women political leaders. They are from different parties in three countries and the aim is to identify the polarity (positive, neutral or negative) of the information published about them in the media. The text focuses on the cases of Anne Hidalgo and Marine Le Pen, from France, Nicola Sturgeon and Theresa May, from the United Kingdom and Ada Colau and Inés Arrimadas, from Spain. The study develops a computerised sentiment analysis of the information published in two leading digital newspapers in each country, during the month of November 2019. The research, with the analysis of 1100 journalistic pieces, shows that the polarity or valence of the women analysed is predominantly neutral and positive and that the journalistic genres do not determine the media representation of the women studied. On the contrary, the country of study does have a predominant incidence on the way in which women politicians are represented, while the relationship of affinity or antipathy of the Spanish media with the women politicians studied is significant.

**Keywords:** women politicians; media representation; sentiment analysis; Spain; France; United Kingdom; gender

## 1. Introduction

Historically, women holding public office or participating in political processes at all levels have been treated differently from their male counterparts. Women politicians have received treatment linked to gender stereotypes and their role in the political life of European societies has been considered secondary (Aaldering and Van Der Pas 2018; Haraldsson and Wängnerud 2019). Women who have a high level of public responsibility have been questioned because of their gender and criticised for aspects that go beyond their political management. These elements have characterised a coverage with a predominantly negative tone that has, at the same time, contributed to the construction of an additional obstacle for women's access to politics on equal terms with men (Bystrom and Dimitrova 2014; Johnstonbaugh 2018; Quevedo and Berrocal 2018).

Despite this, women in different European countries have positioned themselves as important cases of political leadership and have assumed the highest positions of responsibility as prime ministers, mayors of major European capitals or leaders of political parties. Thus, the trend towards the elimination of barriers to women's entry into politics and positions of responsibility has also been driven by more equal coverage in the media (Hayes and Lawless 2016).

The study proposes a systematic and exploratory analysis of the treatment that different media in Spain, France and the United Kingdom give to six women political leaders from different parties: Ada Colau and Inés Arrimadas (Spain), Anne Hidalgo and Marine

Le Pen (France) and Nicola Sturgeon and Theresa May (United Kingdom). Thus, it contributes to obtaining more relevant data on the polarity of the journalistic texts that refer to the specific cases of the women politicians under study. In doing so, this research provides new and original information on the issue of journalistic polarity, which has been less studied (as noted later in the Theoretical Framework) than other issues related to gender and media coverage.

The research aims to identify and analyse the polarity (positive, neutral or negative) of the information published about the cases under analysis in representative and ideologically distant media of the observed countries, in order to characterise the validity of the journalistic coverage of each of them. To achieve the objective, the article implements a quantitative methodology operationalised through the technique of computerised sentiment analysis on 1100 journalistic pieces collected in November 2019. Thanks to the identification of the polarity of the units of analysis identified on the six women studied, it is possible to answer the following three research questions:

RQ1. What is the predominant polarity of media coverage of women politicians in the analysed media?

RQ2. Does journalistic genre determine the valence of women politicians' representation in the analysed media?

RQ3. Do the country and the ideological positioning of the media determine the validity of the representation of women politicians in the analysed media?

It is therefore a relevant study for advancing knowledge on the media representation of women politicians and the polarity of journalistic texts, as it allows us to quantitatively measure issues related to valence, journalistic genre and media countries and ideologies in relation to the specific case studies, which is the main motivation of the research.

The data obtained in the study constitute its main contribution and yield relevant conclusions about the media treatment of the cases studied. It shows a trend of change in the media coverage of women politicians in terms of polarity. The women studied receive, in general, a neutral and positive treatment in all the media and countries studied. However, specific differences can be found in each of the countries or media observed that determine higher levels of negative polarity about some of the women politicians in the sample, defined by contextual ideological structures.

## 2. Theoretical Framework

Women politicians receive different media treatment than their male counterparts. In fact, media coverage of women politicians has been studied from the perspective of gender bias with the aim of observing the existence of inequalities in the representations found in the press. According to Suárez Romero (2017, p. 26), "female underrepresentation is embodied in a triple dimension: that of the media, that of public opinion and politics itself". This is echoed by Aaldering and Van Der Pas (2018) when they explain that the gender differentiation of media coverage contributes to the underrepresentation of women politicians and by Haraldsson and Wängnerud (2019) when they find that, although the overall position of women relative to men has progressed, it does so more slowly in media representation and in politics, as well as by stating that the higher the level of sexism in the media, the lower the proportion of women candidates. In this regard, Barnes et al. (2016), found that less than ten per cent of heads of state were women in the world and they held about twenty per cent of seats in parliaments across the globe. Media coverage of politicians can therefore have electoral consequences given the media's strong influence on society. The media fulfil the agenda-setting function (McCombs and Shaw 1972) and can give more prominence to one or other candidates or issues, thus potentially influencing voters' choices. Testing the existence or not of gender bias in the press representation of male and female candidates is essential given the relationship between media coverage and electoral success or failure (Goldenberg and Traugott 1987; Kahn 1996; Skulley 2017).

While gender discrimination in media coverage of women politicians was notorious in the past, some research suggests that this is no longer the case (Hayes and Lawless 2016).

According to Bystrom and Dimitrova (2014), women were often portrayed by emphasising traditional gender roles, focusing on appearance or their supposed unworthiness to be candidates. For Banwart et al. (2003), however, what is found in today's media coverage are more subtle forms of gender discrimination. According to Johnstonbaugh (2018), the underrepresentation of women in the media is due to their underrepresentation in public office and she notes that the probably more complicated challenge is to get more women into those positions. In contrast, for Courtney et al. (2020) it cannot be assumed that media coverage will increase proportionately as women advance in public life.

Media representations of women politicians are often characterised using stereotypes and so-called trap roles (Quevedo and Berrocal 2018). The "trap roles" (Kanter 1977, pp. 233–36) are four: the seductress, the mother, the mascot and the iron maiden. Over time, Norris (1997) noted that instead of using such stark stereotypes, coverage had shifted towards the use of finer framing: the framing of first woman, a pioneer whose leadership is presented as an advance for all women, and as anomalous (Fernández 2010); the framing of outsider or newcomer character (in which women's abilities to hold office or be candidates are underestimated); the framing of leaders as agents of change (called to make great transformations). For Panke (2015, p. 77), the images of political women according to media coverage are threefold: the warrior (leader and tough), the mother (pending and sensitive) and the professional (hardworking and subordinate). From an empirical point of view, Ríos (2017, p. 65) highlights that the use of "first names, prioritising aspects of private life, understanding the political position in terms of male dependence or substantivising the political position in metaphorical-reproductive terms are some of the trends that predominate in the way in which the media project and help to build a political image of women". Similarly, García Beaudoux et al. (2018) found four recurrent stereotypes: the one that emphasises the role of mother and aspects of domestic life; the one that explains the political careers, achievements and merits of female candidates in relation to powerful or influential men; the one that alludes to the lack of control, rationality or emotional intelligence of female candidates; the one that emphasises the importance of the physical appearance and dress of the female candidate.

O'Neill et al. (2016) have researched the ways in which women politicians are constructed as "other" against the norm of male politicians and find that media coverage emphasises their appearance or femininity; however, less research has been done on women's visibility in the media. For the UK case, Ross et al. (2013) found that women appeared less in the news, and when they did, it was more often in gender-related feature stories, more because of their sex than their political skills or experience. In this vein, studying the 2017 UK election, Harmer and Southern (2019) found that media coverage was dominated by men. Comparing media coverage between British Prime Ministers Margaret Thatcher and Theresa May, Williams (2020) observed that more attention was paid to May's gender and in more detail, particularly in the conservative press. In the French case, Achin and Lévêque (2017) consider that presidential elections are a key moment for the politicisation of sex and gender issues and that this can be observed in media coverage. Looking at the case of Marine Le Pen, Snipes and Mudde (2020) found that the frame of media coverage that predominated was that of the populist radical right, as opposed to a softer gender frame. From an intersectional perspective, Galy-Badenas and Gray (2020) found that mainstream French newspapers reinforce traditional gender roles, and that coverage of female politicians of Maghrebi origin was also affected by their social identities, indicating that there are factors other than gender that can affect media coverage. In this sense, in the Spanish case, Fernández-Garcia (2016) argues that the media coverage received by male and female ministers is different, but that gender is not the only explanation. The distribution of ministries between the sexes also seems to have an influence. Verge and Pastor (2018), for their part, argue that in Spain gendered media frames are omnipresent and that this can lead to the annihilation of the symbolic representation of women.

Van der Pas and Aaldering (2020) found that gender bias exists in the amount of coverage of politicians in proportional electoral systems, in which women receive less

media attention, but not in majoritarian electoral systems. Similarly, the authors confirm that women politicians receive more attention regarding their appearance and personal lives, more coverage of negative viability and, to some extent, coverage of stereotypical issues and traits. Van der Pas and Aaldering's (2020) findings suggest that different types of electoral systems and their characteristics may influence media coverage of women politicians. Other studies also point out that the political slant of the media may condition the portrayal of women politicians depending on the party to which they belong. For Shor (2019), there is mild support for the idea that relatively liberal newspapers are more likely to cover women's issues in a positive light. Indeed, the political affiliation of the media outlet may be one of the factors that helps to understand the differences between media coverage of male and female politicians, but Shor et al. (2014) found that both conservative and liberal newspapers are more likely to cover men.

Regarding the tone of media coverage, the existing literature mostly uses the categories "positive", "negative" and "neutral" to observe the evaluative tone of the news (Zunino 2016), as this factor provides a fundamental evaluative component of the information provided (Sheafer 2007). According to the asymmetry bias theory (Soroka and McAdams 2015; Jacobs and Van der Linden 2018), because of different psychological factors, negative information has a deeper impact than positive information. Thus, the positive or negative tone of news coverage related to women politicians can have electoral visibility and influence, and even professional and personal consequences for its protagonists. Moreover, this is a less analysed element and there is less evidence of it in studies on the media and women politicians in Spain, France and the United Kingdom.

## 3. Material and Methods

The study on the media tone used in news or articles about women politicians is developed through an exploratory methodology that is operationalised in natural language processing (NLP) (Eisenstein 2019; Sun et al. 2019; Wolf et al. 2019; Chowdhary 2020) to describe the type of treatment (positive, neutral or negative) received by the subjects analysed in the media of different countries. This is applied to a corpus of textual analysis with the method of sentiment analysis (Boukes et al. 2020; Arcila-Calderón et al. 2016; Bakshi et al. 2016; Nasukawa and Yi 2003).

Sentiment analysis has been recognised in different studies as a technique to identify the polarity (positive, neutral or negative) of a large set of texts from programming techniques. In the study of communication discourses, sentiment analysis has been mostly used for the identification of polarity in discussion in social media such as Twitter (Kumar and Jaiswal 2020) or Facebook (Sandoval-Almazan and Valle-Cruz 2020; Haryanto et al. 2019) and has also been employed in characterising the polarity of mainstream media discourses (Boukes et al. 2020; Backfried and Shalunts 2016; Padmaja et al. 2014; Balahur and Steinberger 2009; Godbole et al. 2007). Similarly, studies by Leavy (2019, 2020) and Jia et al. (2016) have employed such techniques in the research of media portrayal of women.

In this article, sentiment analysis is used to determine whether the media's treatment of women politicians is positive, neutral or negative and whether there are significant differences between countries and between different media within the same country.

In the study, a sentiment analysis has been carried out on six European women politicians representing three countries: the United Kingdom, France and Spain. The countries chosen have been identified in previous studies (Vliegenthart et al. 2016) as having similar characteristics in the structure of their democracies, which have been defined, in the European context, as majority democracies. At the same time, the choice of the sample of countries responds to the fact that they are similar in terms of size and political relevance in Europe (Alonso-Muñoz and Casero-Ripollés 2020).

For each country, two women were elected who, on the one hand, were or had recently been leaders of relevant political parties in their respective countries, with the condition that each of them represented ideologically opposed political parties. At the same time, all

of them hold or had held very important political positions in their respective countries (leaders of major national parties and members of parliaments, prime ministers or mayors of major cities).

Each of them was studied from all the journalistic pieces found in the period of one month (from 1 to 30 November 2019) in two media of their respective countries defined as ideologically opposed.

The choice of the month studied is relevant in terms of the political events that took place in each of the countries in the sample. In the case of the United Kingdom, it was the month prior to the general election held on 12 December, in which both Theresa May and Nicola Sturgeon played a very important role. In the case of Spain, general elections were held in November 2019. In these elections, both Ada Colau and Inés Arrimadas, leaders of their respective parties, had a significant political and media impact. In the French case, although there were no elections between October and December 2019, there were two historic social protests (the yellow waistcoats and the pensioners) that marked the media and political agenda and determined the political positions of all French political actors, among which Marie Le Pen and Anne Hidalgo were relevant.

The analysis sample is described in Table 1, below:

**Table 1.** Analysis sample.

| Country | Female Politician Analysed | Media Analysed | Journalistic Pieces |
|---------|---------------------------|----------------|---------------------|
| Spain | Ada Colau: Mayor of Barcelona and president of Barcelona en Comú, a left-wing party. | El País | 69 |
| | | ABC | 43 |
| | Inés Arrimadas: Member of the Spanish Congress and president of Ciudadanos, a liberal party. | El País | 63 |
| | | ABC | 54 |
| France | Anne Hidalgo: Mayor of Paris and member of the Socialist Party. | Le Figaro | 23 |
| | | Le Monde | 23 |
| | Marine Le Pen: Member of the National Assembly and president of National Rally, a conservative nationalist party. | Le Figaro | 39 |
| | | Le Monde | 24 |
| Great Britain | Theresa May: Former prime minister and former Conservative and Unionist party leader. | The Guardian | 185 |
| | | The Times | 278 |
| | Nicola Sturgeon: Chief Minister of Scotland and leader of the Scottish National Party (nationalist and social democratic). | The Guardian | 106 |
| | | The Times | 193 |
| Total | | | 1100 |

The study analyses a total of 1100 articles on the six women politicians. The journalistic pieces were collected through a simple search in the FACTIVA software in each of the selected media. For each woman, her full name (first and last name) was used as search words. The selected media were chosen based on two criteria. Firstly, the importance of the media in the national context in terms of reach and circulation. In each country, according to the official circulation indexes, we have chosen the generalist, non-free media with the largest circulation that meet the second condition, described below.

Secondly, that they represented distant ideological positions (conservatives or liberals, according to their own definitions). The selection of the media sample was intended to allow for a comparison of media tone given to women in two media outlets of different political positions within a specific territorial context: Spain (El País, considered a liberal newspaper; and ABC, conservative), France (Le Monde, liberal, and Le Figaro, conservative) and the United Kingdom (The Guardian, liberal and The Times, conservative).

Once the journalistic pieces were found and stored, two cleaning procedures were carried out: firstly, duplicates were eliminated and, secondly, their format was adapted so that they could be entered into a database suitable for content analysis. The figure of 1100 journalistic pieces analysed is based on the final database once the duplicated results were eliminated. In the database, each journalistic piece was categorised under the genre: information or opinion, and each of them was considered as an independent unit of study.

The sentiment analysis process of the MeaningCloud platform (Arcila-Calderón et al. 2016; Kritikos et al. 2020) integrated as an extension within the Microsoft Excel software was

applied to the databases created, one for each political woman studied. Thus, the valence or polarity of positive, neutral or negative sentiment was obtained for each journalistic piece, both informative and opinion genres. The average degree of confidence of the sentiment analysis, according to the values of the study, was 86.1, on a scale of 0–100.

The results of the processing carried out through sentiment analysis allow us to account for the following three variables:

1. Polarity (positive, neutral or negative) obtained by each political woman and by each analysed media according to the percentage of the studied news.
2. Polarity obtained according to the genre of the journalistic pieces discriminated by media.
3. Polarity obtained according to the country studied and the ideology of the media outlet which presents coverage of the women in the sample.

The data obtained allow for a description of the polarity of the journalistic pieces that talk about the women analysed to characterise the tone used by the media when dealing with women politicians. At the same time, it is possible to determine two types of comparison on the mentioned variables. Firstly, comparisons between the media by country are presented (according to the close or distant ideology about the political woman analysed). Secondly, it is possible to compare the results between the different media and countries, which also provides information about the media system itself and its national characteristics of informative treatment of the women politicians investigated.

## 4. Results

In a general analysis, the data obtained in the execution of the sentiment analysis indicate that women politicians have a fewer negative media treatment than is usually thought. Out of the total of the analysed journalistic pieces (1100), 226 (20.5%) indicate a negative polarity, 431 (39.1) a positive polarity and 438 (39.8%) a neutral polarity. Only two articles (0.18%) were identified as very positive, and three texts (0.27%) were not identified with any sentiment. The total results of the sample indicate that the polarity of the media representation of the women analysed is mostly neutral and positive (78.9%), while negative feelings are only manifested in one out of every five journalistic texts observed.

In total, the sample yielded 911 informative pieces and 189 opinion pieces. The difference between the polarity of information and opinion is reflected in Table 2:

**Table 2.** Polarity of media representation of women politicians by journalistic genre [1].

| Polarity | Information | Opinion | Total |
| --- | --- | --- | --- |
| Negative | 175 (19.2%) | 51 (26.9%) | 226 (20.5%) |
| Neutral | 356 (39%) | 82 (42.3%) | 438 (39.8%) |
| Unidentified | 2 (0.2%) | 1 (0.5%) | 3 (0.2%) |
| Positive | 376 (41.2%) | 55 (29.1%) | 431 (39.1%) |
| Very positive | 2 (0.2%) | 0 | 2 (0.1%) |
| Total | 911 (100%) | 189 (100%) | 1100 (100%) |

[1] The percentage, in parentheses, is calculated as a function of the column total to make it more representative.

Table 2 shows that there is a greater negative polarity in the opinion genres than in the information genres, with a difference of 7.7 percentage points, although there is greater neutrality in the opinion pieces than in the informative ones. The biggest difference is found in the positive and very positive polarity of the informative genres, which is more than 12% higher than that of opinion. Even so, it is worth noting that in neither of the two classifications of journalistic genres (information and opinion) does the representation of a negative valence of the female politicians analysed in the total sample have a majority incidence.

In a comparative perspective, by country, the results show greater differences than those found with the journalistic genres. As can be seen in detail in Table 3, Spain is the country with the highest percentage of journalistic pieces with negative polarity (27%),

while France (14.6%) and the United Kingdom (19.4%) are below the average of the study. Likewise, the positive or very positive valence also determines distances between the countries studied. In the first case, France has the best indexes with 61.4% of journalistic pieces with a positive polarity, while the United Kingdom (38.1%) and Spain (31.8%) are distanced by 23.3% and 29.6%, respectively. At the same time, and despite being insignificant for the sample (only two texts), it is important to note that Spain is the only country that does not have texts that are classified as very positive within the sentiment analysis carried out.

**Table 3.** Polarity of media representation of women politicians by country [1].

| Polarity | Spain | France | United Kingdom | Total |
|---|---|---|---|---|
| Negative | 62 (27%) | 16 (14.6%) | 148 (19.4%) | 226 (20.5%) |
| Neutral | 93 (40.6%) | 23 (21.1%) | 322 (42.2%) | 438 (39.8%) |
| Unidentified | 1 (0.4%) | 2 (1.8%) | 0 (0%) | 3 (0.2%) |
| Positive | 73 (31.8%) | 67 (61.4%) | 291 (38.1%) | 431 (39.1%) |
| Very positive | 0 (0%) | 1 (0.9%) | 1 (0.1%) | 2 (0.1%) |
| Grand total | 229 (100%) | 109 (100%) | 762 (100%) | 1100 (100%) |

[1] The percentage, in parentheses, is calculated as a function of the column total to make it more representative.

Neutral polarity within the analysis by country is a few points above the average in both Spain (40.6%) and the United Kingdom (42.2%), and in France, it falls to 21.1%, a figure that places it almost half below the levels of the other two countries.

The disaggregated analyses by country and media also determine differentiated results, which are presented below.

### 4.1. Spain: The Media Tone Is Defined by the Position of the Medium

The sentiment analysis data related to Ada Colau and Inés Arrimadas in the coverage of the newspapers ABC and El País show significant differences in how each media treats one or the other female politician. Table 4 shows opposite behaviours between the two media. In general, Ada Colau receives more negative coverage compared to Inés Arrimadas, but it is notorious that in each media the politicians receive a differentiated treatment. While the negative polarity of Ada Colau in ABC is 37.2%, that of Inés Arrimadas is 20.4% and, at the same time, the positive polarity of Colau in El País is 39.1%, while that of Arrimadas in the same medium is 25.4%. Another noteworthy point is the difference in the neutral ratings. While for Arrimadas the figure reaches 44.4% of journalistic pieces, for Ada Colau it drops almost 10 points, to 36.6%. Neutrality is a factor that remains stable, despite the differentiated values for each female politician, in the two media for each of them.

**Table 4.** Polarity of the media representation of Ada Colau and Inés Arrimadas in ABC and El País [1].

| | Ada Colau | | | Inés Arrimadas | | |
|---|---|---|---|---|---|---|
| Polarity | ABC | El País | Total | ABC | El País | Total |
| Negative | 37.2% | 24.6% | 29.5% | 20.4% | 28.6% | 24.8% |
| Neutral | 37.2% | 36.2% | 36.6% | 42.6% | 46.0% | 44.4% |
| Unidentified | 2.3% | 0.0% | 0.9% | 0.0% | 0.0% | 0.0% |
| Positive | 23.3% | 39.1% | 33.0% | 37.0% | 25.4% | 30.8% |

[1] The percentage, in parentheses, is calculated as a function of the column total to make it more representative.

The polarity analysed in terms of journalistic genres increases the distance between the two media analysed. In the case of Ada Colau, her media treatment in the opinion genre reaches 50%, being the only majority case of negative treatment of female politicians in the entire sample analysed. In the opposite case, Colau only obtains 11% of positive treatment in ABC, also in the opinion genre. In the case of El País, the opinion expressed does not have a negative polarity and the pieces with informative genre reach 40%, 8 points more than the representation of Colau by ABC (32%).

In the case of Inés Arrimadas, the neutral polarity is more predominant, both in opinion and information, in the two newspapers studied. In any case, it is also notorious the negative valence of the case of opinion in the newspaper El País, over 42%, a figure that is close to the negative valuation in the opinion genre of the newspaper ABC (40%). The distance of the negative treatment in the news genres is also significant. In the case of El País, the data show a negative rating of 26.8%, while in ABC this indicator is reduced to 15.9%, with a difference of almost 11 percentage points. On the opposite shore, the positive polarity, there is also a relevant distance. Arrimadas obtains almost 16 points more in ABC (40.9%) than in El País (25%).

### 4.2. France: The Most Positive Treatment for Women

The French case stands out for being the one that represents the most positive polarity when the women analysed appear in the information collected in the study. In the general scenario, we can see how the percentage of information with positive valence is the highest in the sample. Anne Hidalgo stands out with a positive polarity in 76.1% of the journalistic pieces analysed. Although the treatment received by Marine Le Pen does not have such high values and, in fact, marks a difference with Hidalgo of more than 25%, she receives the second-highest positive valence among the six women analysed (50.8%). In the French case, there is a clear difference in the tone used by the two media about each studied woman, without there being, therefore, a political alignment of the media with a specific politician. The data in Table 5 shows a similar behaviour on the levels of difference between each valence for the two women politicians studied.

**Table 5.** Polarity of the media representation of Anne Hidalgo and Marine Le Pen in Le Figaro and Le Monde [1].

|  | Anne Hidalgo | | | Marine Le Pen | | |
|---|---|---|---|---|---|---|
| **Polarity** | **Le Figaro** | **Le Monde** | **Total** | **Le Figaro** | **Le Monde** | **Total** |
| Negative | 4.3% | 8.7% | 6.5% | 17.9% | 25.0% | 20.6% |
| Neutral | 8.7% | 17.4% | 13.0% | 23.1% | 33.3% | 27.0% |
| Unidentified | 4.3% | 0.0% | 2.2% | 2.6% | 0.0% | 1.6% |
| Positive | 82.6% | 69.6% | 76.1% | 56.4% | 41.7% | 50.8% |
| Very positive | 0.0% | 4.3% | 2.2% | 0.0% | 0.0% | 0.0% |

[1] The percentage, in parentheses, is calculated as a function of the column total to make it more representative.

Table 5 also indicates that the percentage of information with a positive valence is much higher in the case of Le Figaro than in the case of Le Monde, for the two women politicians studied. Neutral and negative values are, on the contrary, higher in Le Monde but, as already mentioned, the differences between the two media remain similar for each polarity in the two cases observed. It is also relevant that in both media the positive evaluations decrease in the case of Le Pen and the neutral and, above all, the negative treatments of this politician increase, as shown in Table 6.

**Table 6.** Difference in the polarity of the media representation of Anne Hidalgo and Marine Le Pen in Le Figaro and Le Monde [1].

| **Polarity** | **Le Figaro** | **Le Monde** | **Total** |
|---|---|---|---|
| Negative | 13.6% | 16.3% | 14.1% |
| Neutral | 14.4% | 15.9% | 13.9% |
| Positive | −26.2% | −27.9% | −25.3% |
| Very positive | 0.0% | −4.3% | −2.2% |

[1] The calculation of the difference is always done using the formula difference = Le Pen values − Hidalgo values, corresponding to Table 5.

Anne Hidalgo has a high positive media treatment, and she is the only female politician to achieve, for a specific medium (Le Monde), a very positive rating for almost 5% of

the information collected. For Hidalgo, in the period of analysis, no opinion articles were found in Le Monde and only one in Le Figaro. This single article is the one that determines a slight percentage of negative polarity in the total observed for Le Figaro, as there is no information in this medium that represents Anne Hidalgo with a negative valence. The positive polarity of the mayor of Paris is 86.4%.

In Le Monde, Hidalgo does obtain some negative representation (no more than 9%) and the number of neutral pieces also increases (17.4% compared to 9.1% in Le Figaro), but the positive valence also predominates with 69.6%. Anne Hidalgo's assessment is mostly positive in the two media studied.

The case of Marine Le Pen is different from that of Hidalgo although, as already mentioned, positive evaluations predominate in both media. In Le Figaro, opinion does appear for Le Pen, although she maintains almost the same percentages of negative evaluations as the information genre, 16.7% and 18.2%, respectively. The important differences between genres appear in the neutral polarity, which increases in opinion from 21.2% to 33.3%, and in the positive polarity, which decreases from information (57.6%) to opinion (50%). The total for Marine Le Pen shows a positive polarity of over 56% in the articles analysed in Le Figaro.

The tone used for Marine Le Pen in Le Monde is reduced to the information genre, as no opinion articles referring to her were found in the period studied. In this case, results show the increase of negative and neutral evaluations and, in contrast, a significant drop (−27.9%) in the positive polarity, while the information that presents the female politician with a very positive perspective disappears.

### 4.3. United Kingdom: Women's Position Defines Coverage

In the case of the journalistic coverage by The Guardian and The Times of the politicians Theresa May and Nicola Sturgeon, two differentiated results can be seen according to the female politician analysed. In the case of Theresa May, although the values of negative polarity are not high and are similar in the two media, the main difference lies in the positive valence. While the information with positive polarity is around 30% in the case of The Guardian, for The Times it exceeds 40%. The difference is also reflected in the neutral information estimate, which is almost 50% for The Guardian and 41% for The Times.

In the case of Nicola Sturgeon, as can be seen in Table 7, the similarity of results in the two media analysed is striking. There are no significant differences in any of the polarities (positive, neutral or negative), and it is in neutrality where the greatest distance is found, but it does not exceed two percentage points. Sturgeon reaches an important value in her positive valence (41.8%), six points higher than May and more than ten points ahead of Arrimadas (30.8%), although below French female politicians, the best valued in the analysis carried out.

**Table 7.** Polarity of media portrayal of Theresa May and Nicola Sturgeon in The Guardian and The Times [1].

| Polarity | Theresa May | | | Nicola Sturgeon | | |
| --- | --- | --- | --- | --- | --- | --- |
| | The Guardian | The Times | Total | The Guardian | The Times | Total |
| Negative | 21.6% | 18.7% | 19.9% | 18.9% | 18.7% | 18.7% |
| Neutral | 49.2% | 41.0% | 44.3% | 37.7% | 39.9% | 39.1% |
| Positive | 29.2% | 40.3% | 35.9% | 42.5% | 41.5% | 41.8% |
| Very positive | 0.0% | 0.0% | 0.0% | 0.9% | 0.0% | 0.3% |

[1] The percentage, in parentheses, is calculated as a function of the column total to make it more representative.

Detailed results according to journalistic genre by media outlet show that The Times maintains a consistency in the percentage of news and opinion pieces according to their polarity when either Theresa May or Nicola Sturgeon is mentioned. In fact, the treatment

given to the two women in the two journalistic genres is quite similar to the general values already discussed.

In the case of The Guardian, the incidence of the opinion genre is representative when analysing the tone used for Theresa May. The opinion distances from the average values and the information genre. In opinion, the increase in May's negative polarity stands out above all, reaching 28.9%. There is also an increase in neutral values (55.6%) and a decrease in positive valence (15.6%), the second-lowest value obtained by the female politicians analysed in the sample, which is only ahead of the positive tone given to Ada Colau in the ABC newspaper (11.1%).

In the case of Nicola Sturgeon, The Guardian and The Times treat women politicians homogeneously, with no significant differences depending on the genre analysed. The data allow identifying similar values for each genre.

## 5. Discussion

The study developed confirms the importance of the evaluative tone of the news (Zunino 2016) by finding that most of the journalistic pieces analysed from all countries contain some assessment (positive or negative). The tone of the pieces studied shows that the negative polarity is lower than what might be expected based on the results of previous studies that examined the differential treatment of women politicians in the press. Thus, the preponderance of a polarity, in general, mostly neutral or positive of the studied women politicians deepens in the change of tendency towards a less discriminatory media coverage towards women who participate in politics and occupy relevant public positions. In this way, the presence of more subtle forms of gender discrimination, such as those studied by Banwart et al. (2003), tends to diminish in the media representation of women, and the results obtained in the exploration of content in terms of its valence suggest that the negative tone related to the representation of women is becoming less frequent and the focus on women is placed on diverse factors, and not only on gender.

Despite the lower presence of women in public office compared to men (Barnes et al. 2016), the study sample shows that several women have held or hold relevant representative positions in the countries analysed. All the women analysed, despite being a small group, head their political parties, are recognised leaders and are highly represented in the media.

Regarding media coverage and its possible influence on electoral success, the fact of finding a mostly positive or neutral polarity towards women in the media analysed suggests that the possibility of accessing public office is increasingly higher for women, as shown by the cases of Anne Hidalgo and Ada Colau, who have revalidated their positions as mayors of Paris and Barcelona, respectively. The cases of Inés Arrimadas, Marine Le Pen and Nicola Sturgeon, as leaders of their parties, are also representative and are indicative of the fact that women's own media exposure has allowed them to obtain or maintain their own leadership, and to be protagonists in the media studied.

Contrary to what Johnstonbaugh (2018) argues, the six women analysed account for an advance both in the media representation of women and in the occupation of relevant public positions. Under the theory of asymmetry bias (Soroka and McAdams 2015; Jacobs and Van der Linden 2018), it can be inferred that more positive coverage of women politicians could give them greater visibility and influence, as there is less negative news or articles, which, although they could have a strong impact, would be in the minority and could see their effects counteracted by the greater amount of positive or neutral media coverage.

However, there is still a manifest inequality in public life between men and women and it cannot be assumed that a more positive or higher media treatment, as a single factor, automatically leads to a higher female presence in politics (Courtney et al. 2020). The detailed study of the correlation between a higher presence—and positive treatment—of women in the media and their access to relevant public office is a relevant research question for future research.

With respect to the Spanish case, following Shor (2019), it is found that the political affiliation of the media can be a factor that determines the tone of the journalistic coverage with respect to the women politicians analysed. In Spain, Colau (left-wing leader) receives a more negative tone in the conservative and monarchist newspaper, ABC, compared to Arrimadas, a liberal leader. In the same line, El País (considered a progressive media) has a higher percentage of information with negative polarity of Arrimadas than of Colau. This trend, however, is not repeated in the other media analysed, which maintain a greater independence in their coverage of the female politicians in the sample.

In the French media studied, the high positive polarity found (the highest in the sample) follows the line of previous studies (Snipes and Mudde 2020) in the fact of not representing the women analysed in a negative way. This is particularly the case for Marine Le Pen, where the positive tone was found even in the most progressive media, Le Monde. In France, moreover, the two studied media present the same tendency of coverage of the two analysed women politicians, without their ideological or political position (both of the media and of the politician) determining a more positive or negative evaluation in the tone of the media treatment carried out.

In contrast to previous research, the results for the UK denote consistent coverage of May and Sturgeon, which is not focused on gender but on their role as high-level political representatives. This is seen in the large number of both news and opinion pieces appearing for these two cases, which also gives them greater visibility.

On a more ideological level, contrary to the findings of Shor's (2019) studies, which show a slight tendency for more liberal media to be more likely to represent women's issues positively in the media, the study data indicate that the highest percentages of positive polarity are found in newspapers with more conservative tendencies, except for the Spanish case. Thus, in France, Le Figaro has a higher positive representation of both Hidalgo and Le Pen than Le Monde and, in the United Kingdom, The Times has a much higher positive representation of May than The Guardian and a small difference in the positive coverage of Sturgeon, which is one percentage point higher in The Guardian. In the Spanish case, the partisan and polarised trend is maintained and while El País has a more positive coverage of Colau, with a difference of more than 15% compared to ABC, the conservative newspaper has an 11.6% more positive assessment of Arrimadas than El País.

## 6. Conclusions

The sentiment analysis conducted on the media coverage of six female politicians in three European countries yields specific and comparative data on the treatment of female politicians in the media. Firstly, the study shows that the polarity or valence of the women analysed (Ada Colau, Inés Arrimadas, Anne Hidalgo, Marine Le Pen, Theresa May and Nicola Sturgeon) is not negative. On the contrary, the predominant valence of the women analysed is positive or neutral. Both values exceed the percentages assigned to negative polarities and in none of the six cases studied is the negative valence higher than the positive and neutral. This finding is one of the most relevant of the study, within its limited scope, because it is different from previous research results found in the literature where the negative approach was predominant.

Secondly, the data collected show that the journalistic genre does not determine the tone (positive, neutral or negative) used in the coverage of the women analysed. In general, the incidence of journalistic genre in the treatment of women is scarce, except for the Spanish newspapers (El País and ABC) that mark, in the opinion about Inés Arrimadas and Ada Colau, respectively, the only negative representations that in the disaggregated detail exceed the neutral or positive polarity.

The previous results give rise to the third conclusion related to the media treatment of women in the countries and, also, in the specific media. In the case of the sample studied, the country of study does have a greater impact on the way in which women politicians are treated in the media. While in Spain the distance and the indirectly proportional relation between the media and its representation of the two women politicians studied

is evident, under a very well-defined ideological and polarised position; in France and the United Kingdom the representation of the women studied are less differentiated and keep similar characteristics or tendencies when women politicians belonging to distant political parties are represented. In the same way, the French case is distinguished, apart from having the most positive evaluation of women—contrary to Spain, which has the most negative ones—because there is no comparison or alignment between a media and a female politician. On the contrary, the two French media studied behave in a similar way when comparing the two women in Le Figaro and Le Monde.

The case of the United Kingdom, which is closer to the French case, behaves independently of the favourable or unfavourable alignment towards Theresa May or Nicola Sturgeon. The data trajectories indicate homogeneity in the treatment, especially in the case of Sturgeon. The only deviation found in the homogeneous trend is seen in the opinion texts in The Guardian, where May's negative polarity increases a little more and breaks the equal behaviour of the obtained representations.

The study shows that women politicians receive differentiated treatment according to the media and political culture of each of the countries observed. Countries with high rates of polarisation, such as Spain, tend to represent in their media the same political tension of the daily political scenario. In this sense, the comparison of Spain with the United Kingdom, but above all with France, reflects an important contrast on the political distance between a media outlet and the politicians represented in its contents. Nevertheless, to achieve deeper generalisations, further studies and different methodological techniques would be needed in addition to those used in this study.

Likewise, although it does not compare with the treatment given to male politicians, the analysis yields important data on the progress in the recognition of women as political actors who receive a high level of media attention and that their journalistic reflection, as part of the political system, maintains a mostly neutral or positive polarity. Future studies can use sentiment analysis in conjunction with detailed information on each particular case of media coverage to identify the specific treatments and tones that each media outlet uses to portray women politicians. At the same time, a future line of research could be to conduct longitudinal, comparative and trend studies on constant analysis samples to determine whether differences persist in the media's treatment of women politicians compared to men or whether, on the contrary, the struggle for women's media equality has borne some valuable fruit in recent years.

**Author Contributions:** Conceptualization, C.F.-R. and S.G.-L.; methodology, C.F.-R. and S.G.-L.; software, C.F.-R. and S.G.-L.; validation, C.F.-R. and S.G.-L.; formal analysis, C.F.-R. and S.G.-L.; investigation, C.F.-R. and S.G.-L.; resources, C.F.-R. and S.G.-L.; data curation, C.F.-R. and S.G.-L.; writing—original draft preparation, C.F.-R. and S.G.-L.; writing—review and editing, C.F.-R. and S.G.-L.; visualization, C.F.-R. and S.G.-L.; supervision, C.F.-R. and S.G.-L.; project administration, C.F.-R. and S.G.-L.; funding acquisition, C.F.-R. and S.G.-L. All authors have read and agreed to the published version of the manuscript.

**Funding:** This research received no external funding.

**Institutional Review Board Statement:** Not applicable.

**Informed Consent Statement:** Not applicable.

**Data Availability Statement:** The data presented in this study are available on request from the corresponding author.

**Conflicts of Interest:** The authors declare no conflict of interest.

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
