# Peer review of "How Are Women Politicians Treated in the Press? The Case of Spain, France and the United Kingdom"

_journalmedia, doi:10.3390/journalmedia2040043_

Round 1

Reviewer 1 Report

Thank you for your interesting, correct and good researches

Author Response

Dear Reviewer 1,

Thank you for your kind comment. We are very pleased that you found our work to be adequate. We appreciate the time you have taken to review our work.

Reviewer 2 Report

The article is an interesting comparison of media coverage of conservative and liberal party leaders in three countries. This article makes an important contribution by controlling for ideology. The methodology is appropriate and the conclusions are relevant to the findings. I would like to see a little more explanation of a few details. First, I would like the author to justify the selection of female cases only as opposed to male-female matched cases. We don't know if media coverage of women politicians is more negative or positive than men of their parties. 

Second, I would like to see a justification for selection November 2019 as the month of analysis, a little description of the salient issues on the agenda, and the roles the women play in their parties. For instance, May was no longer a party leader but LePen is. 

Finally, the author should play up the principle finding that women received relatively little negative media coverage, which stands in opposition to previous findings. I suggest making this point first in the conclusion and work backwards to the less significant findings. 

Author Response

Dear Reviewer 2,

Thank you for your suggestions for improvement. We appreciate the time you have taken to review our work. Following your comments, we have better justified the selection of the cases of women politicians, as well as the period of analysis and the situation of the cases under analysis, together with the most relevant media issues of the moment. We have added information to justify the choice of the whole sample, based on objective data and previous literature.

Likewise, we have emphasised, in the conclusions, the main findings in order to clarify the contributions of the article.

We hope that with the changes we have made, the article will gain in clarity and quality.

Reviewer 3 Report

Overall, I think this paper takes on an important and relevant topic. I also think the data collection is interesting, systematic and shows a lot of potential. But as I will outline below, I also identify several weaknesses in the paper which all together leads me to the decision to reject the paper.

Introduction

There is neither any mentioning nor motivation of the research contribution the paper makes. I’ve read the section ‘Introduction several times and is still not able to detect a trace of motivation for what explicit research (theoretical and/or empirical) contribution the paper makes, what gaps in previous research it has identified and how the current research aims to fill this gap. This further leads to that it is hard to assess the chosen research questions. Why are these three RQ’ the most important ones? How are they connected to the (currently unidentified) research contribution? Finally, I also find it very strange that the first ten sentences in the paper are without any references to previous research, despite all the different claims stated in these sentences.

Theoretical framework

The theoretical framework feels rushed. The most part is devoted to discussing previous research on different gender stereotypes connected to the media representation of woman politicians. Although I think this is important research, it’s immediate connection to the research questions in the paper is weak. Of course, it’s important to provide a wider literature review than what is explicitly connected to the RQ’s, but the consequence is that the review of previous research that is connected to the three RQ’s (polarity, journalistic genre, country and ideological position of the media outlet) receives less attention and becomes superficial. The framework would have benefitted from a less detailed description about gendered stereotypes in the representation of woman politicians, for more discussion about previous research closer connected to the research questions.

Material and methods

My main concerns with the paper are connected to case selection and the design of the study. Recognizing the challenges of doing comparative content analysis of media, there are several things design-wise neither mentioned nor discussed and this affects the generalizability of the results.

First, why are these three countries chosen? I’m sure there are good reasons, that may or may not be related to electoral system and media market structure (as shortly discussed in the literature review), but there are no explicit arguments for this selection.

Second, why are these specific six politicians selected? I lack a discussion about the selection of woman politicians and perhaps also what possible women politicians that you choose not to include. For all countries, one politician has a local or regional attachment, but they still vary a lot. Ada Colau is a mayor/local level politician, but she is also a mayor in the largest city in Catalonia, where there has been a heated conflict about independence for years. To put it careful, it’s likely this conflict also reflects on how she is portrayed in the Spanish media founded in Madrid. In Great Britain, the regional dimension is represented by Nicola Sturgeon, but she is also a chief minister, a very powerful and prominent position, but as a representative for the SNP. Thus, there are similar issues to take into consideration as in the Spanish/Catalan case. For the UK case, you also don’t have any equivalence to the mayors of Barcelona or Par. Furthermore, you choose Teresa May who is a member of parliament, but a former prime minister and party leader.

Third, there is no description of the chosen media outlets, neither a more general description (related to their position on the national media market), nor any information about their ideological positions, despite this being mentioned as a key reason for selecting these media outlets. The media outlets’ ideological position should also be explicitly connected to the party/ideology of the studied politician, as there seems to be an implicit argument that this might affect how the politician is portrayed.

Fourth, the chosen time period (November 2019) should be coupled with information about whether there was any election or election campaign going on in the different countries or not. For sure, the news reports in UK and Spain would have been more characterized by election campaign coverage as these both countries had elections to the national parliament this month and year.

All in all, the motivation for, and arguments behind, the design and the case selection must be made much more explicit. What were the strategic decisions regarding case selection and variation in the various aspects (countries/media markets, media outlets, politicians, time period)? It’s both about disclosing necessary information about several crucial aspects as detailed above, but also to explicitly connect the comparative case design with how far the obtained results could travel. This said, the data collection is both impressive and seem very systematic.

Results

Overall, the empirical section is fairly straightforward, but at the same time, there are so many possible ways to cut the results section. Although the results section aligns with the research questions, it’s not entirely clear why you compare countries for some aspects but not for others (for example, genre). The variable ideological position sort of disappears. While this is explicitly mentioned as part of RQ3, it is not one of the three variables mentioned in methods/material, and it’s only implicitly dealt with in the results section (but there is a short mentioning of this in the Discussion). A table distinguishing media representation of woman politicians between liberal/conservative outlets would have been helpful. Finally, it’s unclear why there is a table comparing polarity in the French case and not in the other cases.

Discussion and conclusion

The discussion and the concluding section suffer from that there is no initial research gap identified and no strong contribution stated. It’s hard to truly assess the results, especially as the main comparison the authors seem to be interested in, gender discrimination, need a comparison with man politicians to be able to say something more robust. The result that journalistic genre only matters in Spain, without contextualizing this finding more than saying it’s because Spain is a polarized country, feels strange. I also don’t think it’s possible, based on the study, to draw general conclusions about differences across countries or the increasing possibilities for women of accessing public office. This said, the discussion and the conclusion provide answers to the three research questions initially stated.

Author Response

Dear Reviewer 3,

We appreciate your time and effort in reviewing our work. We understand your comments on the weaknesses of the article, and following your suggestions, we have tried to improve the different sections, especially concerning the methodology. 

In the beginning, we have added references that justify some parts of the introduction, which are then detailed in the Theoretical Framework, and we have better justified the motivation and contributions of the article. We hope this makes the connection to the research questions clearer. In reference to the Theoretical Framework, it was written precisely in this way to highlight the gap in knowledge on the topic of polarity and journalistic texts in the case of media representation of women politicians, as opposed to other topics (e.g. gender stereotypes). In this sense it has been tried to clarify that the contribution of the article is related to providing specific data to the debate on polarity and gender in journalistic texts. 

In the section explaining the method, we have better explained the design the study design, as well as the choice of cases, countries, media and time period analysed. This has been done on the basis of previous literature and objective data and their respective sources. In this way, justifications have been added for each item.

Moreover, all the comparisons made have been reviewed and the variables have been clarified, especially with regard to ideological position. At the same time, some generalisations of the findings have been qualified, as the limitations of the study do not yet allow for categorical statements.

In addition, the English has been revised. We hope that the article has gained some clarity and improved in its weaker parts with the changes made.